# Effect of Light Characteristics on the Sensory Properties of Red Lettuce (*Lactuca sativa*)

**DOI:** 10.3390/foods10112660

**Published:** 2021-11-02

**Authors:** Kelly Gude, Martin Talavera, Audra M. Sasse, Cary L. Rivard, Eleni Pliakoni

**Affiliations:** 1Department of Horticulture & Natural Resources, Kansas State University, 22201 West Innovation Drive, Olathe, KS 66061, USA; kellygude@ksu.edu; 2Department of Food, Nutrition, Dietetics and Health, Kansas State University, 22201 West Innovation Drive, Olathe, KS 66061, USA; talavera@ksu.edu (M.T.); audrasasse@ksu.edu (A.M.S.); 3Department of Horticulture & Natural Resources, Kansas State University, 35230 West 135th Street, Olathe, KS 66061, USA; crivard@ksu.edu

**Keywords:** sensory lexicon, visible light, ultra-violet light, hoop-house

## Abstract

Leafy green production in high tunnels (HTs) results in increased yields, improved visual quality, and extended production with polyethylene (poly) film and/or shade cloth coverings. However, altering visible and ultra-violet light with HT coverings may reduce phytochemicals, thus influencing plant pigmentation and taste. The objective of this study was to examine various HT coverings on the sensory perceptions, soil temperature, color, and anthocyanin accumulation of red leaf lettuce. The coverings included standard poly, standard poly with removal two weeks prior to harvest (movable), diffuse poly, clear poly, UV-A/B blocking poly (block), standard poly with 55% shade cloth, and the open field. A highly trained descriptive panel evaluated the samples using a scale from 0 (none) to 15 (extremely high) and determined a list of 20 sensory attributes. The color intensity attribute had the most differentiation between coverings, and the open field was higher (i.e., darker) than the others at 7.5 (*p* < 0.0001), followed by clear and movable coverings at 6.8, and the shade covering scored a 2. Strong relationships existed between both colorimetric (hue°) and anthocyanin analysis to panelist-based scores (R^2^ = 0.847 and 0.640, respectively). The initial crispness was similar for movable, standard, diffuse, and block coverings at 5.3 on average, which was higher than the open field at 4 (*p* < 0.01). The open field lettuce grew under cooler soil temperatures, which may have slowed down maturation and resulted in softer tissue. Based on this study, HT growers can implement specific coverings to cater to markets that value visual quality.

## 1. Introduction

Fruit and vegetable consumption may lead to the reduction of certain diseases, due to their antioxidant properties [1,2]. The phenolic compounds are the largest category of phytochemicals that act as antioxidants scavenging free radicals, neutralizing them to prevent lipid oxidation [3,4]. Phenolic acids (such as caffeic acid) and flavonoids (such as anthocyanin and quercetin) act as antioxidants pre- and post-consumption in both plant tissues and consumers. In humans, they have been shown to reduce arteriosclerotic plaques and inflammation [5], and habitual consumption is associated with decreased mortality due to cardiovascular and cancer [6]. In plants, the phenolic compounds provide pigmentation and defend against pest attacks with a bitter flavor and astringent chemesthetic sensation [7,8,9].

Lettuce, especially red leaf lettuce, is a good source of phenolic acids and flavonoids, including quercetin glucosides, anthocyanin conjugates, and caffeic acid derivatives [10,11,12,13,14,15]. Amongst the environmental factors that affect crop phytochemical quality, light characteristics (both solar intensity and spectral quality) are particularly important [16,17]. Light-induced production of flavonoids and phenolic acids in the outer tissues of plants provides a well-known protective mechanism against intense solar radiation [18,19]. Sensory properties are very important for the assessment of vegetable quality by consumers and for their purchase behavior. The red pigment in lettuce leaves is due to phenolic compound accumulation, especially anthocyanin, and is an important appearance attribute responsible for the commercial value of red leaf lettuce [20].

With the recent growth in local food production, there has been an expansion of high tunnel (HT) systems [21]. HTs are utilized by growers for environmental protection, increased marketability, and an extended production season [22,23]. HTs may be a more accessible and cost-effective option for lettuce production as compared to a greenhouse [24]. The most common cool-season crop grown in HTs is lettuce [21]. Typically, HTs are covered with 6-mil polyethylene (poly) film that is ultra-violet- (UV) stabilized for durability. However, the HT system allows for a grower to select particular poly films and/or shade cloth in order to affect the spectral quality of light, microclimate, and crop growth [25], as well as phenolic compounds [16,25,26].

Looking specifically at natural light sources, UV light obtained from solar radiation is a large factor affecting certain phenolic compound accumulation and subsequent plant pigmentation and appearance [11,16,27,28,29,30,31]. It was found that the red leaf lettuce grown under full solar UV-radiation had a deeper red pigmentation compared to the other coverings with partial UV transmission [31]. In agreement, it was found in an environmentally controlled study that UV-B stimulated the biosynthesis of anthocyanin and antioxidant polyphenols [27]. It has been reported [29] that both solar radiation and temperature have a positive correlation with color and the phenolic acid and flavonoid content of three red leaf lettuces. It was found that plant pigmentation positively correlated with total phenolic content and increased as the season progressed [29]. In partial agreement, it was suggested that low plant temperature exerted a stronger positive influence on anthocyanin and red pigmentation in lettuce compared to light [20].

There have been a few studies that report the effects of HT vegetable production on sensory quality [32,33,34]. Researchers have examined pac choi grown with varying levels of conventional and organic fertilizer in both HT and open field production systems [34]. A trained panel determined that pac choi from within the HT had higher intensities of most attributes (umami, crispness, sulfur, green overall, or woody), regardless of the fertilization management. Others yet studied the detectable differences between store-bought spinach and locally grown spinach from both the HTs and the open field [32]. They observed that the overall likeness, flavor, and texture of HT grown spinach was preferred to open field grown or store-bought spinach in both a large consumer study and a descriptive panel. 

This study examines the individual sensory attributes of the red lettuce grown under different HT coverings and compares them to soil temperature, leaf color, and anthocyanin concentration. By identifying and evaluating these lettuce attributes, we can inform growers on the ways in which light quality under HTs affects lettuce quality. This study is a portion of a larger study that researched the effect of HT coverings on microclimate, productivity, phytochemical accumulation, and now- sensory attributes. The first objective of this study was to develop sensory profiles by a highly trained descriptive panel to describe color, flavor, texture, and mouthfeel characteristics of red leaf lettuce to determine if any differences exist when grown under different HTs coverings. The second objective was to quantify the effects of the coverings on microclimate (soil temperature), leaf color, and anthocyanin concentration, and test relationships to the perceived panelist score of color intensity.

## 2. Materials and Methods

Trials were conducted at the Kansas State University Olathe Horticulture Center, located in Olathe, Kansas. The trial was conducted in four “caterpillar” HTs that were 39.6 m long × 3.7 m wide × 2.1 m high. The construction of a homemade HT allowed for customization to suit the needs of the experiment regarding plot size and tunnel length. The overall design of the caterpillar tunnel is long and narrow with low ceilings, which provides an ideal structure for an experiment that specifically examines the impact of solar light. Each replication (high tunnel) had a total of six consecutive covering plots (6.1 m long) that spanned the width of the tunnel (3.7 m wide) and were joined at the long end by hip boards. An additional 2.1 m buffer area was included at the ends of each tunnel as well as 1.5 m at either end of each plot to minimize interplot interference. Data were collected from the middle 3.1 m of each plot. The lettuce trials utilized a split-plot randomized complete block design (RCBD), blocked by HT. Two beds ran lengthwise in each HT (39.6 m long × 0.61 m wide), and the lettuce varieties, ‘New Red Fire’ and ‘Two Star’ (Johnny’s Selected Seeds, Winslow, AZ, USA), alternated between the north and south bed at every other tunnel (tunnels and rows were altered from east to west). For the purpose of this study, only the red variety, ‘New Red Fire’, will be discussed. Lettuce was seeded 7 September 2017, into 72-cell propagation trays (4 cm diameter) (Pro-Tray 72 Cell Flats; Johnny’s Selected Seeds, Winslow, ME, USA) with potting mix and transplanted four weeks later (6 October 2017). Common cultural methods for the region were practiced during production [35]. Lettuce was transplanted in a staggered double row within the bed (26.7 cm between plants, 26.7 cm between rows). Water was applied through drip irrigation. The trial was irrigated three days per week (Monday, Wednesday, Friday) for 60 and 90-min per irrigation event.

Each tunnel, or rep, included six individual plots that were randomly assigned to the six coverings. An open field (open) bed was added adjacent to the HTs, but the replicates were not randomized and used for comparison purposes. The six HT coverings included commercially available greenhouse and HT poly films as well as shade cloth. The standard poly (standard) was rated for 92% PAR transmission and blocked <350 nm [single-layer 6-mil (K-50 poly; Klerk’s Plastic Product Manufacturing, Inc., Richburg, SC, USA)]. Standard poly + removal (movable) allowed for plant establishment in a semi-protected environment before full solar exposure 2 weeks prior to harvest. The movable covering simulated the potential use of a movable tunnel [25]. Diffuse poly (diffuse) removed direct radiation of infra-red (IR) light and blocked <380 nm (Luminance; Visqueen Building Products, London, UK). Clear poly (clear) did not contain a UV-inhibitor (6-mil Clear Plastic Sheeting; Lowes, Mooresville, NC, USA). UV-A/B block poly (block) blocked <400 nm (Dura Film Super 4; BWI Companies, Inc., Nash, TX, USA). A 55% shade cloth + standard poly underneath (shade) reduced plant temperature (Sunblocker Knitted Shade; FarmTek, Dyersville, IA, USA). The HT coverings, codes, and expanded descriptions are seen in Table 1.

### 2.1. Lettuce Sampling

Once the lettuce was mature within the HTs, plants were harvested the morning of 7 November from each plot, using a lettuce knife (Harris Seeds, Rochester, NY, USA) at the soil level to remove the full plant along with any outer whirl leaves, minus the root system.

For the descriptive study, two lettuce plants were chosen at random from each covering plot, within the four replications, bagged, and boxed in coolers of ice for transport to the testing facilities in Manhattan, KS for sensory evaluation in an air-conditioned vehicle. This study used the methods described in the manuscript, “Lexicon to Describe Flavor of Fresh Leafy Vegetables” [38]. The study was conducted in six days: two days for orientation and lexicon development, and the remaining four days for evaluation of the samples in triplicate (90 min for each day).

For both instrumental (colorimetric) measurements and spectrophotometric (anthocyanin) analysis, six plants were chosen randomly from each covering plot, within the four replications, placed in plastic bags, and transported in an air-conditioned vehicle to the postharvest physiology lab at KSU-Olathe. Analysis occurred on the day of harvest and after 5 days of storage in optimum conditions of 1.5 °C and 90% RH in environmental chambers (Forma Environmental Chambers; ThermoFisher Scientific Inc., Asheville, NC, USA). These time points were chosen to account for any changes that would occur during the time required for the descriptive lexicon development and evaluation stages.

### 2.2. Lexicon Development and Evaluation Procedure

A highly trained descriptive panel (*n* = 5) composed of four females and one male between 58 and 77 years of age from the Center for Sensory Analysis and Consumer Behavior at Kansas State University (Manhattan, KS, USA) evaluated the samples. These panelists completed at least 120 h of training and had a minimum of 2000 h of sensory testing experience. More specifically, panelists had previous experience in the evaluation of vegetables and other vegetable products [32,38]. During the development of the lexicon, the panelists were asked to examine terms that had previously been developed to describe the sensory attributes of a wide variety of green vegetables, including lettuce [38], as well as terms used to describe “green” aroma in foods [38]. Starting from this previous work, the existing lexicons were adjusted to only include terms applicable to lettuce. The same panelists have been used as well as the previously mentioned lexicons with minor adjustments for fresh spinach [32]. The work with spinach helped the panel in the development of the lexicon used in this study with lettuce; both studies used fresh, leafy vegetables, and the lexicons were mostly similar. Development sessions lasted 90 min, and up to six samples were evaluated in each session. Several products were reviewed during the development phase to adjust the lexicon and familiarize the panel with the product category. Table 2 shows a complete list of the attributes used for evaluation, including the attribute definitions, references, and intensities for attributes not obtained from [38].

The evaluation method used was adapted from the flavor profile method [39,40]. The flavor profile method uses a panel consensus in which the panelists must come to an agreement on definitions, attributes, and reference products in the development of the lexicon. In this study, sample evaluation was executed by panelists individually and in triplicate, using a 0–15 intensity scale with 0.5 increments, with 0 meaning “none” and 15 meaning “extremely high”, and was compliant with ASTM standards [41].

The red lettuce samples were evaluated after two days in closed plastic bags in refrigerated storage (4 °C). For preparation, each lettuce sample was rinsed with deionized water and dried using a salad spinner. Random, similar leaves with no evidence of deterioration were chosen for evaluation (whole leaves without petiole removal). Each replicate for each lettuce covering was served in sample sizes of four to six leaves, and three replications were utilized in total. The leaves were served on 4-inch foam plates with random 3-digit codes to reduce bias. If needed, leaves were cut to fit onto the plates. Each panelist received one random leaf. To test the lettuce, panelists were instructed to fold the leaf in half through the middle and take one bite from the middle of the fold. The panel room had neutral colors, was well lit, ventilated, temperature-controlled, and it was compliant with ASTM standards [42].

### 2.3. Soil Temperature

Various aspects of the microclimate (UV, PAR, and soil and canopy temperature) were observed in the trials, and details are reported in Gude [25]. HT soil and canopy temperatures (°C) were continuously recorded with two probes (EL-USB-1; Lascar electronics, Erie, PA, USA) per plot; the soil probe was buried 10 cm below the soil surface, and the canopy probe was at the soil surface, similar to [43]. The probes were placed in the north row of the HTs in the center of each covering plot. All sensors were connected to a programmable data logger to record temperature in 30-min increments, and the results are the average of the minimum and maximum temperatures. Sensors collected temperature from 27 October to 7 November 2017 (12 days). For the purpose of this study, we will focus on soil temperatures, as the canopy temperature differences don’t support or dissuade the results.

### 2.4. Color

The color was measured on each of the four replicates of each of the covering treatments, comprised of three lettuce heads, using an A5 Chroma-Meter (Minolta CR-400; Minolta Co. Ltd., Osaka, Japan). Two measurements were taken on the left and right sides of the midrib on an undamaged outermost leaf, 1 to 3 cm from the tip. Color results were expressed by the chromatic coordinates CIE L*, a*, b*, hue, and Chroma [44]. Following color measurements, full-head samples were combined by replications, lyophilized in the freeze dryer (Harvest Right, Salt Lake City, UT, USA), and ground (Waring WSG30; Conair Corporation, Torrington, CT, USA) for anthocyanin analysis.

### 2.5. Anthocyanin Extraction and Measurement

To extract and measure anthocyanin, each of the four replicates of the covering treatments was comprised of three full lettuce heads and was extracted and analyzed in a darkened room with a red safety light to avoid photo-oxidation, following a previous procedure [45]. Lyophilized lettuce (0.2 g) was homogenized with 4 mL of extraction solution (ethanol/water, 80/20, *v*/*v*) (VWR, Radnor, PA, USA), vortexed (20 s), sonicated (5 min) (Ultrasonic Bath; Fisher Scientific, Hampton, NH, USA), and centrifuged (4000 rpm, 15 min, 4 °C) (Avanti J-E; Beckman Coulter, Indianapolis, IN, USA). The supernatant was transferred into a test tube, and the extraction was repeated. Both supernatants were combined and evaporated to dryness under nitrogen flow (2–6 ppm) and recovered with 4 mL of 30 mM ammonium acetate (VWR, Radnor, PA, USA) in de-ionized (D.I.) water with 5 pH adjusted with formic acid (VWR, Radnor, PA, USA) and filtered through a 25 mm 0.22 μm filter (Supor; VWR, Radnor, PA, USA) into several 1.5 mL Eppendorf tubes for reserve and the sample extract was stored in darkness at −70 °C until analysis.

Prior to analysis, a portion of each extracted sample was thawed, vortexed and 150 µL sample extract was pipetted in triplicate on 96-well microplates. Absorbance in the microplate reader with the spectrophotometer (Biotek Synergy H1MDl; BioTek Instruments, Inc., Winooski, VT, USA) was measured at 530 nm to correspond with anthocyanin pigments in red lettuce [46]. Finally, standard curves were developed using HPLC grade cyanidin 3-glucoside (Acros Organics, Geel, Belgium) from 0.39063 to 50 µg/mL. Results were expressed as µg/g dry weight (DW).

### 2.6. Statistical Analysis

Analysis of variance (ANOVA) and Fisher’s LSD (Least Significant Difference) were conducted on the dataset to determine significant differences between coverings. The fixed effects of the 3-way ANOVA model were covering, panelist, and replication. Multivariate analysis was also done in the form of a principal component analysis (PCA) and cluster analysis using the Ward method to assess distances and to further compare relationships between covering groups. The analysis was performed using XLSTAT 2018.5.52459 (Addinsoft, New York, NY, USA).

The soil temperature, lettuce color, and anthocyanin response parameters were analyzed under the linear mixed model. Fixed effects of the model were covering and analysis day (lettuce color and anthocyanin). The random effect of the model is HT. Pairwise comparisons between coverings were performed based on the 2-sided test for non-zero difference in means. The adjustment for multiplicity was carried out using Tukey’s method at the 0.05 significance level. When there was no statistical difference between the sampling days, as in the case of the lettuce color, and anthocyanin concentration, the values were combined between analysis days. The analysis was performed using JMP Software (JMP Pro 14.1.0; Cary, NC, USA).

## 3. Results and Discussion

A lexicon of 20 attributes (Table 2) was developed to describe the appearance, texture, flavor, and mouthfeel characteristics of the red lettuce. Significant differences were found in eight attributes (color intensity, initial crispness, water-like, toothetch, parsley, woody, sweet overall, and astringent) (Table 3). Full sensory profiles were generated for each covering.

Color intensity was the attribute that had the most differentiation between samples grown under different coverings as compared to the other attributes (*p* < 0.0001; Table 3). The shade covering was significantly lower in color intensity compared to the other coverings. The open field scored significantly higher than the other coverings but was followed closely by the clear and movable coverings. The open field had full sunlight exposure throughout the growth process, the movable covering plots were fully exposed two weeks prior to harvest, and the clear covering transmitted both UV-A and UV-B. Clear and movable coverings maximized light exposure with photosynthetic active radiation (PAR) at 85 and 100% transmission, respectively, and available UV-A/B at 61/65% and 100% transmission, respectively (Table 1). Previously, it has been shown that the amount of light exposure a plant receives during the growth process has the greatest impact on pigmentation [17,31,47,48]. It was found that UV-radiation is a greater determinate on the coloration of red lettuce than overall light intensity [31]. Specifically, plants were significantly redder under 100% UV-radiation and 100% light in the open field, compared to plants in a 0% UV-radiation and 100% visible light environment. Covering treatments were also distinguishable by colorimetry and spectrophotometric analysis as L*, a*, chroma, hue angle, and anthocyanin concentration differed significantly among coverings (Table 4). The open field and movable covering (both with 100% UV and 100% PAR; Table 1) and clear covering had lower L* values (i.e., more black), higher a* values (i.e., redder), and lower hue and chroma values (i.e., darker in color; data not shown) compared to the shade covering.

The texture characteristic of initial crispness scored similarly between movable, standard, block, and diffuse coverings and greater than the open field (*p* < 0.001; Table 3). In the present study, the soil temperature of the open field was cooler at an average of 10.4 °C compared to all other coverings (*p* < 0.0001). It has been reported that immature lettuce leaves have softer tissue [49], which may have contributed to the perceived lack of crispness of the open field by the sensory panel. Increased temperatures (over 12 °C) speed up maturation [37,50] and therefore, yields [25,43,51]. Besides the open field, the other tested coverings had soil temperatures of 12 to 13 °C, meaning that the lettuce in the open field may have had an earlier maturity stage at harvest. The shorter HT structures provided a good method for conducting replicated studies focusing on light. Because the plots were adjacent to each other, there is an opportunity for interplot interference as it relates to temperature and relative humidity. Careful management of the tunnels helped to minimize this phenomenon as sidewalls and endwalls were open during daytime hours to allow for ample ventilation. Future studies on a much larger scale could include individual HTs for each plot (*n* = 24) to better focus on the effect of covering on temperature.

Regarding flavor, the “green” flavors in this study included green-overall, green-peapod, green-grassy/leafy, and green-viney. Of the green flavors listed, the mean intensity scores of each covering were low to moderate and not significant (Table 3). In addition to the “green” flavors, three common leafy vegetables were used as a reference on the scale for comparison: lettuce, spinach, and parsley. The samples scored low on spinach and parsley, and moderately on lettuce flavor. For parsley flavor, it was noted that clear and standard coverings scored high in comparison to open field and shade coverings (*p* < 0.05). The flavor attributes of woody and musty/earthy scored low for all coverings. However, the woody attribute was high for the diffuse covering in comparison to open, clear, and movable coverings. Other terminologies, such as sweet-overall, sour, bitter, salty, and umami, were used to describe the taste. All lettuce samples scored very low for all of these attributes, except for the “bitter” category, where it scored moderately. However, no significant differences were noted between coverings, except for overall sweetness. Diffuse scored significantly higher than standard, and shade coverings and open field.

For the mouthfeel characteristic, water-like, the standard covering scored statistically higher than all other coverings (Table 3). For toothetch, all samples scored low, and the open, movable, standard, and shade coverings were statistically similar and greater than the block covering. Mouthfeel characteristics, water-like, and toothetch, were used to describe products that possessed “green” attributes [38,52], along with astringent and metallic mouthfeel. For both astringent and metallic, all lettuce coverings scored low. However, some differences were noted for astringent where the open field provided lettuce with the highest astringency and was statistically greater than the clear and block coverings.

From the descriptive analysis, it was observed that the largest differences between coverings were related to color intensity (*p* < 0.0001), initial crispness (*p* < 0.001), and water-like attributes (*p* < 0.001; Table 3). Those three attributes helped to formulate two clusters of coverings that were evident from the principal component analysis (PCA) and confirmed by cluster analysis under the Ward method (Figure 1). The first cluster included lettuce grown within the open, clear, and movable coverings, this cluster is characterized by lettuce with the highest color intensity, which explains about 73% of variability among coverings by itself. The second cluster was formed by samples from the diffuse, standard, and block coverings. This cluster was characterized by the samples with the highest initial crispness and water-like attributes, which explains 13% of the total variability. These coverings may have a better quality considering lettuce crispness attributes influence consumer perception of product freshness [53]. A single covering, shade, was not grouped into either cluster because it was different from the rest with low color intensity. Flavor attributes did not have a huge effect explaining the differences between coverings.

The lettuce under the shade covering scored lower for color intensity as the light was reduced with shade cloth. Furthermore, the anthocyanin concentration was decreased under the shade compared to the movable covering (Table 4). Studies have shown that decreased UV [14,16] and PAR [17,29,48,54,55] prior to harvest plays a negative role in red pigment and anthocyanin concentration in red lettuce. It was found that red lettuce anthocyanin concentration was 46% higher when grown under no-shade compared to a 50% black shade cloth [55]. In the present study, the shade covering had just 40% PAR and 7% UV-A/B transmission (Table 1).

Furthermore, lower environmental temperatures have been shown to increase plant pigmentation while slowing down plant maturation [20,29,33]. In the present study, the open field had significantly lower soil temperatures, as previously discussed, and scored the highest in color intensity (Table 3). In a baby red leaf lettuce study comparing climatic variables to plant pigment and phenolic content, [29] confirmed that lettuce harvested in February showed a longer growing cycle and was a darker red than its warmer counterpart harvested in May. It is hypothesized that either transcription of the genes involved in the biosynthesis of anthocyanin compounds increases to improve cold temperature tolerance [20,29,56], or that anthocyanin content accumulates due to a lowered rate of photosynthesis at colder temperatures (10 °C) [57]. However, the effect of cold temperatures has been found to be species and cultivar specific in lettuce [20,29]. During this fall trial, the soil temperatures were relatively cool under all coverings, and there was no evident relationship between soil temperature and anthocyanin concentration (R^2^ < 0.1).

Increases in color intensity scores from the panelists were strongly associated with decreases in hue angle (R^2^ = 0.847; Figure 2a) and increases in anthocyanin concentration (R^2^ = 0.640; Figure 2b). Although weaker, associations between anthocyanin values and colorimetric hue angle (R^2^ = 0.507; Figure 2c) were encouraging. Similar to past studies [20], the data show a potential for nondestructive colorimetric-based measurements of leaf redness to serve as an approximation for panelists’ color intensity scores and anthocyanin concentration.

## 4. Conclusions

This study clearly shows that light variations during growth have significant effects on the color intensity of red lettuce, explaining most of the differentiation between coverings. Because consumers purchase based on appearance and prefer redder red leaf lettuce- growers will want to design their HT system to promote leaf pigmentation. From an appearance perspective, open, clear, and movable coverings provided the consumer with a darker red leaf lettuce. However, clear and movable coverings also benefitted from the controlled environment while maintaining more pigmented tissue compared to the lettuce grown under the other coverings. The shade covering differed from the others, as it resulted in lettuce with the least red pigment and the least anthocyanin content. The strong relationship between color measurements and color intensity from the descriptive panel can benefit lettuce breeders looking to incorporate nondestructive procedures into their breeding programs. Explaining a smaller portion of variation between coverings was texture and mouthfeel, as the diffuse, standard, and block coverings had a high initial crispness and water-like attributes. Texture and mouthfeel are important quality parameters for the consumer but explained less variability between coverings when compared to color intensity (13% versus 73%, respectively). There was little difference in the flavor attributes between lettuce coverings. These results indicate that the spectral quality of UV and visible radiation alters perceived texture, mouthfeel, and visual parameters. Since these parameters are known to play an important role in the commercial value of red leaf lettuce, covering materials may be of considerable importance from a color intensity standpoint.

## Figures and Tables

**Figure 1 foods-10-02660-f001:**
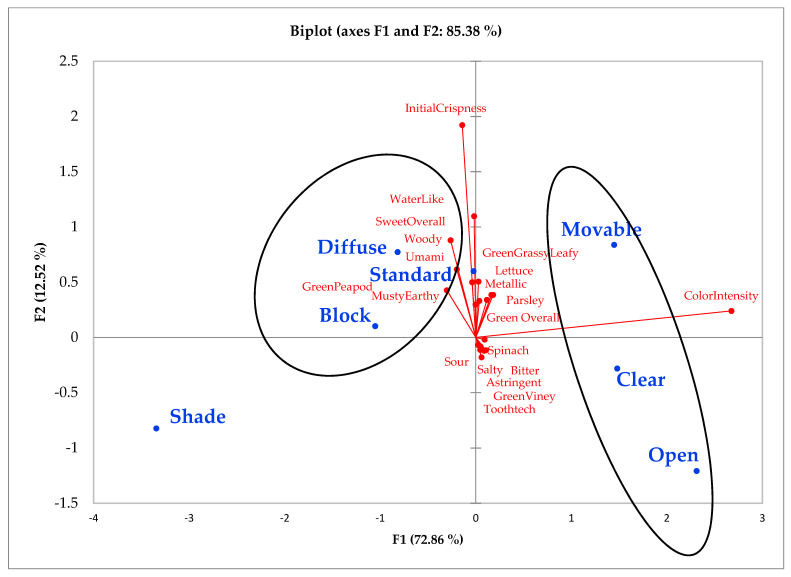
Representation of the polyethylene light treatments with principal component analysis (PCA) map of factor 1 (visual; movable, clear, and open) vs. factor 2 (texture; diffuse, standard, block). Treatment codes: open field (open); clear poly (clear); standard poly with removal two weeks prior to harvest (movable); standard poly (standard); diffuse poly (diffuse); UVA + UVB blocking (block); standard poly with shade cloth (shade).

**Figure 2 foods-10-02660-f002:**
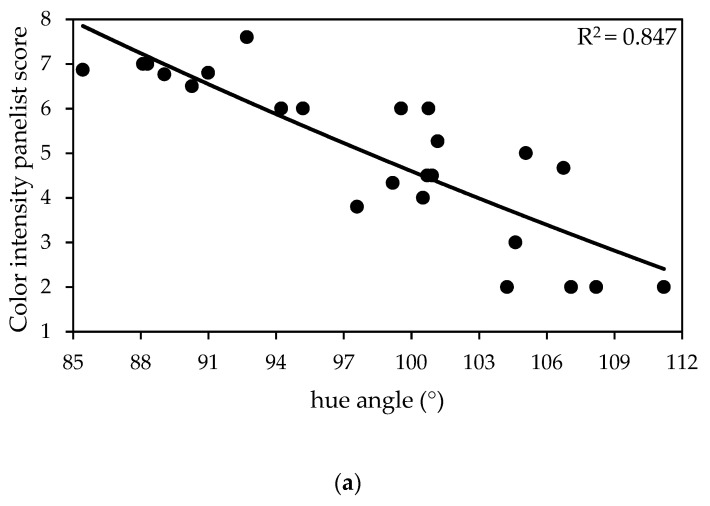
Relationship between: (**a**) Hue angle measured with a colorimeter and panelist color intensity scores (0–15 scale) given by panelists; (**b**) Anthocyanin concentration measured with a spectrophotometer and color intensity scores; and (**c**) Anthocyanin concentration and hue angle for lettuce grown under 6 different high tunnel coverings in Olathe, KS in fall 2017.

**Table 1 foods-10-02660-t001:** Polyethylene (poly) coverings in the high tunnel system, corresponding codes, product descriptions, and ultra-violet (UV) and photosynthetic active radiation (PAR) transmission for the studied lettuce samples.

Covering	Code	Description	UV-A/B Transmission (%) ^1^	PAR Transmission (%) ^2^
Standard poly	Standard	Single layer, 0.15 mm rated for 5-year lifetime.	16/16 a	83 b
Standard poly (removal 2 weeks prior to harvest)	Movable	Allows full spectral light once removed.	100/100	100
Diffuse poly	Diffuse	Removes direct radiation of IR light, reducing leaf temperature, deepens light penetration.	8/7	76
Clear poly	Clear	Allows full spectral light without much filtration, increased rate of degradation because does not contain UV-inhibitor.	61/65	85
UV-A/B Block (380–400 nm)	Block	Slows degradation of plastic.	24/6	84
55% Shade Cloth + Standard	Shade	Reduces light intensity and temperature.	7/5	39
Open Field	Open	Allows full spectral light without filtration.	100/100	100

^1^ Measured with the ILT5000 (International light Tech., Peabody, MA, USA) on cloudless days [36,37]; ^2^ Measured with the CID-340 (CID Bio Science, Inc., Camas, WA, USA) on cloudless days [25].

**Table 2 foods-10-02660-t002:** Sensory categories and attributes used to describe color, flavor and texture of red leaf lettuce.

Category	Attributes	Definition	Reference
Appearance	Color Intensity (Redness)	Intensity or strength of the color from light to dark.	Pantone color chips 2042U = 8.0
Texture	Initial Crispness	The intensity of audible noise at first bite with the molars.	Fresh Baby Spinach Leaf = 2.5 Snow Pea = 8.0 Fresh Spinach—Place 5 in Ziploc Snack bags; Snow Pea—Serve 3 in Ziploc Snack bags; Fold leaf in half from leafy end to stem end. Take bite at center of fold.
Flavor	Green, overall	Aromatic characteristics of plant-based materials. A measurement of the total green characteristics and the degree to which they fit together. Green attributes include one or more of the following: green-unripe, green-peapod, green-grassy/leafy, green-viney, and green-fruity. These may be accompanied by musty/earthy, pungent, astringent, bitter, sweet, sour, floral, beany, minty, and piney.	[38]
	Green, Peapod	A green aromatic associated with green peapods and raw green beans; characterized by increased musty/earthy.	
	Green, Grassy/Leafy	A green Aromatic associated with newly cut-grass and leafy plants; characterized by sweet and pungent character.	
	Green, Viney	A green aromatic associated with green vegetables and newly cut vines and stems; characterized by increased bitter and musty/earthy character.	
	Lettuce	Green, slightly musty and sometimes bitter water-like aromatics associated with lettuce like Bibb and Iceberg.	
	Spinach	The brown, green, slightly musty, earthy aromatics associated with fresh spinach.	
	Parsley	The clean fresh green, bitter, pungent aromatics associated with fresh parsley.	
	Woody	Brown, musty aromatics associated with very fibrous plants and bark.	
	Musty/Earthy	Aromatics associated with damp, wet soil	
	Sweet, Overall	Aromatics associated with the impression of sweet substances such as fruit or flowers.	
	Sour	The fundamental taste sensation of which citric acid is typical.	
	Bitter	A basic taste factor of which caffeine is typical.	
	Salty	The fundamental taste factor of which sodium chloride in water is typical.	
	Umami	Flat, salty flavor enhances naturally occurring in some tomatoes.	
Mouthfeel	Water-like	Liquid perception during mastication of some fruits and vegetables such as watermelon, peaches, tomatoes, and lettuce.	
	Tooth-etch	A chemical feeling factor perceived as drying/dragging when the tongue is rubbed over the back of the tooth surface.	
	Astringent	The drying, puckering sensation on the tongue and other mouth surfaces.	
	Metallic	An aromatic and mouthfeel associated with tin cans or aluminum foil.	

**Table 3 foods-10-02660-t003:** Descriptive analysis attribute mean ^1^ scores and *p*-value of red leaf lettuce grown in high tunnel systems under six different high tunnel coverings or open field.

Treatments ^2^	Color Intensity	Initial Crispness	Green Overall	Green Peapod	Green, Grassy/Leafy
Open	7.5 (0.1) a ^3^	4.0 (0.3) c	5.0 (0.2)	1.9 (0.3)	3.9 (0.2)
Clear	6.8 (0.1) b	4.6 (0.3) bc	5.1 (0.2)	2.2 (0.4)	4.2 (0.3)
Movable	6.8 (0.2) b	5.6 (0.3) a	5.2 (0.2)	2.5 (0.3)	4.3 (0.2)
Standard	5.3 (0.3) c	5.3 (0.2) ab	5.2 (0.2)	2.5 (0.4)	4.5 (0.2)
Diffuse	4.7 (0.3) d	5.5 (0.2) a	5.1 (0.2)	2.5 (0.3)	4.2 (0.3)
Block	4.3 (0.1) d	5.0 (0.2) ab	5.1 (0.2)	2.3 (0.3)	4.0 (0.3)
Shade	2.0 (0.0) e	4.6 (0.3) bc	4.8 (0.2)	2.8 (0.3)	4.1 (0.2)
*p*-value	<0.0001	0.001	ns ^4^	ns	ns
Treatments	Green Viney	Lettuce	Spinach	Parsley	Woody
Open	2.5 (0.2)	4.3 (0.1)	1.8 (0.3)	1.9 (0.2) c	2.2 (0.3) cd
Clear	2.4 (0.4)	4.6 (0.2)	2.0 (0.3)	2.5 (0.2) a	2.2 (0.3) d
Movable	2.2 (0.3)	4.6 (0.1)	1.7 (0.3)	2.1 (0.2) abc	2.3 (0.3) bcd
Standard	2.7 (0.3)	4.4 (0.2)	1.9 (0.3)	2.5 (0.2) ab	2.7 (0.3) ab
Diffuse	2.4 (0.3)	4.5 (0.2)	1.6 (0.3)	2.0 (0.3) bc	2.8 (0.2) a
Block	2.0 (0.3)	4.5 (0.1)	1.9 (0.3)	2.1 (0.2) bc	2.6 (0.2) abc
Shade	2.4 (0.3)	4.1 (0.2)	1.7 (0.3)	1.8 (0.3) c	2.4 (0.3) abcd
*p*-value	ns	ns	ns	0.05	0.05
Treatments	Musty Earthy	Sweet Overall	Sour	Bitter	Salty
Open	2.9 (0.3)	0.3 (0.2) c	0.7 (0.2)	5.9 (0.2)	0.2 (0.1)
Clear	2.4 (0.4)	1.2 (0.3) ab	0.8 (0.2)	5.5 (0.2)	0.5 (0.2)
Movable	3.0 (0.3)	1.1 (0.2) ab	0.5 (0.2)	5.7 (0.2)	0.3 (0.2)
Standard	3.0 (0.4)	0.8 (0.2) bc	1.1 (0.2)	6.1 (0.1)	0.2 (0.1)
Diffuse	3.1 (0.2)	1.8 (0.3) a	0.4 (0.2)	5.5 (0.2)	0.2 (0.1)
Block	2.7 (0.3)	1.3 (0.2) ab	0.5 (0.2)	5.6 (0.2)	0.3 (0.2)
Shade	2.8 (0.4)	1.1 (0.2) b	0.6 (0.2)	5.6 (0.3)	0.3 (0.2)
*p*-value	ns	0.001	ns	ns	ns
Treatments	Umami	Water-like	Tooth-etch	Astringent	Metallic
Open	2.6 (0.2)	3.8 (0.3) bc	2.7 (0.2) a	1.9 (0.2) a	0.3 (0.2)
Clear	3.1 (0.2)	3.6 (0.2) c	2.3 (0.1) bc	1.5 (0.2) bc	0.7 (0.2)
Movable	3.2 (0.2)	4.3 (0.3) b	2.7 (0.2) a	1.7 (0.2) abc	0.7 (0.2)
Standard	2.6 (0.2)	5.1 (0.2) a	2.6 (0.2) ab	1.9 (0.1) ab	0.5 (0.2)
Diffuse	3.1 (0.2)	4.2 (0.2) bc	2.4 (0.2) abc	1.6 (0.2) abc	0.6 (0.2)
Block	3.5 (0.2)	4.1 (0.3) bc	2.3 (0.1) c	1.4 (0.2) c	0.4 (0.2)
Shade	2.8 (0.3)	3.8 (0.3) bc	2.6 (0.1) a	1.6 (0.2) abc	0.5 (0.2)
*p*-value	ns	0.001	0.05	0.05	ns

^1^ Data are LSmean values (SE) (*n* = 5) on a 15-point scale. ^2^ Treatments: open field (open); clear poly (clear); standard poly with removal two weeks prior to harvest (movable); standard poly (standard); diffuse poly (diffuse); UVA + UVB blocking (block); standard poly with shade cloth (shade). ^3^ For each attribute not sharing the same letter within the same column were significantly different at *p* < 0.05 (Fisher’s protected LSD). ^4^ ns is not significant.

**Table 4 foods-10-02660-t004:** Effect of high tunnel covering ^1^ on average soil temperature, color based on L* (−black to +white), A* (−greenness to +redness), and hue angle [tan^−1^ (b*/a*)], and anthocyanin concentration of lettuce grown in Olathe, KS in fall 2017. Within the same column, means ^2,3,4^ with different letters are different (*p* ≤ 0.05), Tukey’s HSD.

Covering	Soil Temp		Color						Anthocyanin Concn. (µg/g DW)	
	°C		L*		A*		hue (°)			
Open	10.4	c	34.8	c	2.5	a	75.3	c	–	
Clear	13.4	ab	37.7	bc	−0.4	ab	89.6	b	1170.62	ab
Movable	12.1	b	38.1	bc	−1.4	b	91.9	b	1272.51	a
Standard	13	ab	38.5	b	−3.5	bc	98.4	ab	965.41	ab
Diffuse	13.5	a	40.7	b	−5.6	cd	102.9	ab	1190.57	ab
Block	13.3	ab	40.3	b	−4.3	bc	100.3	ab	1121.46	ab
Shade	12.9	ab	45.3	a	−8.5	d	107.7	a	609.82	b
*p*-value		<0.0001		<0.0001		<0.0001		<0.0001		<0.05

^1^ Trial was arranged in a RCBD, blocked by high tunnel, with the following 6 covering treatments randomly assigned within each tunnel: clear poly (clear); standard poly with removal two weeks prior to harvest (movable); standard poly (standard); diffuse poly (diffuse); UVA + UVB blocking (block); standard poly with shade cloth (shade). An open field bed (open) was adjacent to the high tunnels and replicated plots were not randomized. ^2^ Soil temperature probes added 10 cm below the soil surface, recording temperature in 30 min increments (2 probes per treatment), from 27 October to 7 November 2017 (12 days). ^3^ Color values are lsmeans of 96 lettuce measurements (4 measurements per plant, 6 plants per rep). ^4^ Anthocyanin values are lsmeans of 24 lettuce plants (6 plants per rep). Anthocyanin concentration of lettuce from the open field was not measured.

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
