# Peer review of "Effect of Light Characteristics on the Sensory Properties of Red Lettuce (Lactuca sativa)"

_foods, 2021, doi:10.3390/foods10112660_

Round 1

Reviewer 1 Report

The paper presents a study on the sensory properties of red lettuce grown in high tunnels under different covering materials and in the open field. Perceived color is also related to measured color and anthocyanin concentration. Soil temperature under the different coverings is shown, too.

In my opinion the most interesting and innovative aspect of the work is the sensory analysis of lettuce, that makes data worth publication. Nevertheless, the paper suffers some lacks, especially in terms of information about methodology and experimental conditions. Discussion is somehow unclear and conclusions are not completely supported by data.

Below are my observations, which I hope will be useful to improve it.

Keywords: I find some keywords not (completely) appropriate. “Lexicon” is a very generic word and generally used in completely different fields, so I suggest, at any rate, “sensory lexicon”; since throughout the article the used system of plant protection is called “high tunnel”, why choosing “hoop-house” as keyword instead of “high tunnel”?; “season extension” is not a suitable keyword since the work does not focus on it at all.

Experimental design:

-Split-plot designs are used for factorial experiments, while in this experiment there is only one factor (covering type), so I am not sure that in this case the experimental design can be defined as “split-plot”;

-The authors should explain how they avoided the influence of the adjacent coverings on a specific area (covered by e certain covering) inside the same tunnel (from the text: “each tunnel, or rep, included six individual plots that were randomly assigned the six coverings”, lines 108-109).

Description of the covering materials:

-In some points of “Results and Discussion”, transmission of PAR and UV by some of the coverings was reported, but not of all of them, and in a quite confused way. I think that it would be very useful for the readers if authors add a column in table 1 with this information, and recall it in the text when necessary. “Movable” is supposed to have the same behavior as “standard” before removal, and 100% transmission after it.

Captions and notes of tables and figures: they need to be more accurate and standardized (same style, same information, etc…).

Order of results presentation/discussion: I think it would be more logic to put data of table 3 before figure 1. Furthermore, in table 3 please put parameters in the order of table 2.

The authors wrote: “Various aspects of the microclimate were observed in the trials and details are reported in Gude (2020)” (line 192-193), but did not specify which aspects. Furthermore, it is never easy for a reader to look for data in another publication. Also, it is not clear to me why the authors chose soil temperature as the only climatic parameter to show in the paper, and only as the average of the period. So please add data about the trend of temperature (both soil and air) and radiation intensity during the trial. This would also help to enhance discussion and will support the sentence (now not really supported by shown data) “This study clearly shows that light and temperature variations during growth have significant effects on color intensity and texture of red lettuce, explaining most of the differentiation between coverings” (lines 385-387).

The authors hypothesize than the lack of crispness of lettuce grown in the open field was due to softer tissue, in turn due to earlier maturity stage, in turn due to low soil temperature. Probably reality is more complex. A factor the was not considered at all is water availability: how water (i.e. irrigation) was managed in the HTs vs. open? Some information should be added in Materials and methods and, if necessary/useful, this aspect should be considered in the discussion.

I suggest to change lines 389-392: from “From an appearance perspective, open, clear and movable coverings provided the consumer a darker red leaf lettuce. However, clear and movable coverings also benefitted from the controlled environment while maintaining more pigmented tissue. The shade covering differed from the others, as it resulted in lettuce with the least red-pigment and the least anthocyanin content. The strong relationship between color measurements and color intensity from the descriptive panel can benefit lettuce breeders looking to incorporate nondestructive procedures into their breeding programs. From a texture and mouthfeel perspective, the diffuse, standard, and block coverings had high initial crispness and water-like attributes which is another important consumer quality”, to “from “From an appearance perspective, the open filed cultivation provided the consumer a darker red leaf lettuce. However, clear and movable coverings benefited from the controlled environment while maintaining more pigmented tissue than the other coverings. On the other hand, from a texture and mouthfeel perspective, the diffuse, standard, and block coverings had high initial crispness and water-like attributes which are other important quality parameters for the consumer. The shade covering differed from all the others, as it resulted in lettuce with the least red-pigment and the least anthocyanin content. The strong relationship between color measurements and color intensity from the descriptive panel can benefit lettuce breeders looking to incorporate nondestructive procedures into their breeding programs”.

Other remarks are reported directly in the manuscript (see attached file)

Author Response

Thank you for taking the time to review the manuscript and add comments throughout the narrative. Please see the attachment

Reviewer 2 Report

The manuscript is very interesting and very well drafted. I only have two minor remarks:

Line 52: The abbreviation "HT" is mentioned for the first time in the main text. Please explain it.

Line 103: Please explain the abbreviation "RCBD".

Author Response

Thank you for taking the time to review the manuscript.

Point 1 - Line 52: The abbreviation "HT" is mentioned for the first time in the main text. Please explain it.

Response 1 - Thank you for your suggestion, this has been altered (line 52).

Point 2 - Line 103: Please explain the abbreviation "RCBD".

Response 2 - Thank you for your suggestion, this has been altered (line 103).

Round 2

Reviewer 1 Report

The authors have improved the manuscript in some aspects, but they have ignored some remarks/suggestions or did not respond satisfactorily.

Anyway, the crucial points is that, from the revised version of the manuscript, is evident that the different covering plots were not separated, which did not allow to avoid the influence of the adjacent coverings/different treatments on a specific cultivated area/treatment. Unfortunately, this is a serious, not solvable, methodological problem that seriously affects the experiment.

Author Response

Thank you for the time that you have spent reviewing this manuscript. The responses to this second review have been taken into account and I have added on to the first review in the attached document. I have revised the manuscript further and have responded to any of your comments from the first review that were not incorporated, but now with a better explanation. Thank you again.
